# Synthesis of Polystyrene-Based Cationic Nanomaterials with Pro-Oxidant Cytotoxic Activity on Etoposide-Resistant Neuroblastoma Cells

**DOI:** 10.3390/nano11040977

**Published:** 2021-04-10

**Authors:** Silvana Alfei, Barbara Marengo, Giulia Elda Valenti, Cinzia Domenicotti

**Affiliations:** 1Department of Pharmacy, University of Genoa, Viale Cembrano, 16148 Genoa, Italy; 2Department of Experimental Medicine (DIMES), University of Genova, Via Alberti L.B., 16132 Genoa, Italy; barbara.marengo@unige.it (B.M.); giuliaelda.valenti@edu.unige.it (G.E.V.); cinzia.domenicotti@unige.it (C.D.)

**Keywords:** antimicrobial cationic polymers, chemoresistance, human neuroblastoma cells, ROS damage

## Abstract

Drug resistance is a multifactorial phenomenon that limits the action of antibiotics and chemotherapeutics. Therefore, it is essential to develop new therapeutic strategies capable of inducing cytotoxic effects circumventing chemoresistance. In this regard, the employment of natural and synthetic cationic peptides and polymers has given satisfactory results both in microbiology, as antibacterial agents, but also in the oncological field, resulting in effective treatment against several tumors, including neuroblastoma (NB). To this end, two polystyrene-based copolymers (P5, P7), containing primary ammonium groups, were herein synthetized and tested on etoposide-sensitive (HTLA-230) and etoposide-resistant (HTLA-ER) NB cells. Both copolymers were water-soluble and showed a positive surface charge due to nitrogen atoms, which resulted in protonation in the whole physiological pH range. Furthermore, P5 and P7 exhibited stability in solution, excellent buffer capacity, and nanosized particles, and they were able to reduce NB cell viability in a concentration-dependent way. Interestingly, a significant increase in reactive oxygen species (ROS) production was observed in both NB cell populations treated with P5 or P7, establishing for both copolymers an unequivocal correlation between cytotoxicity and ROS generation. Therefore, P5 and P7 could be promising template macromolecules for the development of new chemotherapeutic agents able to fight NB chemoresistance.

## 1. Introduction

Resistance to antibiotics or to chemotherapeutic agents is a multifactorial phenomenon that, by limiting the therapeutic efficacy of drugs, has a negative impact on patients’ survival and on health care costs [1,2,3,4]. Therefore, to counteract the development of resistance, research is increasingly focused on developing alternative therapeutic strategies based on the use of natural or synthetic cationic compounds able to exert cytotoxic effects not involving specific molecular mechanisms or enzymatic processes that cells could change and thus acquire a resistant phenotype. In this regard, differently structured cationic materials, including natural and synthetic cationic peptides, synthetic cationic polymers, and dendrimers, by interacting with negative constituents of both Gram-positive and Gram-negative bacteria’s surfaces, induce depolarization and permeabilization of cell walls and membranes, leading to irreparable and lethal damage of the cytoplasmic membrane, loss of cytoplasmic content, and finally to cell death [3,4,5,6,7,8,9]. Although some bacteria such as *Pseudomonas aeruginosa* can actively but temporarily produce cationic peptides able to disguise the negative charge of their outer membrane, thus developing resistance, most bacteria do not develop such a defense system, remaining susceptible to the cytotoxic effects of these materials [10].

Furthermore, another advantage of using such materials is their selectivity toward the prokaryotes cells’ external envelope. In fact, the microbial cell wall—being rich in phosphatidylserine, phosphatidylglycerol, or cardiolipin, as well as teichoic acid residues—is more appealing to cationic molecules [11], while the cytoplasmic membrane of mammalian cells—having fewer negative charges than those of bacteria due to the presence of a higher fraction of neutral lipids, such as phosphatidylcholine, sphingomyelin, and phosphatidylethanolamine [4,11,12,13]—is less sensitive to the cytotoxic action of these materials. Moreover, the presence of cholesterol, which is absent in prokaryotes, confers to the eukaryotic cells major rigidity and improved barrier properties, thus displaying a remarkable protection against the disruptive action of cationic macromolecules [4,11]. Interestingly, it has been found that cancer cells are more sensitive to the attack of cationic compounds than not-transformed eukaryotic cells [2]. This different susceptibility is because, similar to prokaryotic cells, cancer cells possess a strongly negatively charged cell surface due to the overexpression of phosphatidylserine, the presence of *O*-glycosylated mucines, and the incidence of glycosaminoglycan side chains mainly in the form of heparin sulfate [2,14]. Additionally, in comparison with normal cells, the membrane of cancer cells have lower levels of cholesterol (like bacteria), but higher levels of microvilli, improving the possibilities of interaction with cationic macromolecules. Moreover, cancer cells enhance their susceptibility toward these cationic devices by altering their receptor accessibility, cell adhesion, and communications with the extracellular environment [15]. The mechanism through which these cationic macromolecules exert their cytotoxic action on cancer cells are similar to those observed in bacteria. In detail, these cationic compounds, by a simple contact, create irreparable damage to a cell membrane, create permeable channels, and induce mitochondrial damage, as well as inducing other non-membranolytic intracellular actions that eventually lead to caspase-dependent cell apoptotic death [16].

Considering the strong analogy of the cytotoxic mechanism through which these cationic compounds act on tumor and bacterial cells, the identification of new cationic polymers can offer new therapeutic opportunities both in the microbiological and oncological fields.

In the microbiology sector, research is currently focused on using monomers containing primary ammonium groups in the form of acidic salts to find selective and potent novel cationic antibacterial polymers able to mimic the amphiphilic properties and cationic functionalities of antimicrobial peptides containing lysine [3,4]. Indeed, polymers containing primary ammonium salts have been shown to outperform their tertiary and quaternary analogues in terms of more potent antibacterial activity and lower hemolytic toxicity [4]. Moreover, a common practice to soften excessive densities of cationic charge, to tune the balance between hydrophobic and hydrophilic properties, and to control the polymer length, consists of mixing the cationic monomer with uncharged acrylates, methacrylates, acrylamides, and methacrylamides as comonomers to prepare less hemolytic copolymers [4]. In this regard, we recently reported the synthesis, characterization, and remarkable broad spectrum antibacterial activity of a cationic random copolymer (P5) obtained by copolymerizing a cationic 4-ammoniumbuthylstyrene monomer (M5) with uncharged dimethylacrylamide (DMAA) [4] (Scheme 1).

Since P5 was shown to be highly cytotoxic toward prokaryotes cells, we considered that it could also be an excellent candidate as a cationic device for counteracting chemoresistance in cancer cells.

In this study, starting from M5 and P5 as template cationic monomer and macromolecule, a new more complex monomer (M7) was designed, synthetized, and copolymerized with DMAA to prepare a novel random copolymer (P7). In comparison with P5, P7 was characterized by a higher average molecular mass (Mn), a more rigid structure, due to the presence of two phenyl rings in the structure of M7, and the absence of the C4 chain as a spacer between the ammonium group and the phenyl ring. The cytotoxic activity of P5 and P7 was investigated on two human neuroblastoma (NB) cells lines differently sensitive to etoposide (ETO), a chemotherapeutic drug commonly used to treat NB and which exerts its cytotoxic effect by stimulating reactive oxygen species (ROS) production, which induces oxidative stress (OS) in cancer cells.

In this regard, it has been widely documented that oxidation therapy based on ROS production is an effective strategy to induce cytotoxic effects in cancer cells. Recently, this approach has been implemented by using different methodologies, such as the employment of polymeric nanoreactors (NRs) able to improve in vivo pro-oxidant efficiency [17,18]. In detail, it has been reported that glucose oxidase (GOD)-loaded ROS-responsive polyion complex vesicles, when activated by tumor metabolic acidity, were able to stimulate hydrogen peroxide (H_2_O_2_) production, thus increasing tumor redox state. At the same time, high levels of H_2_O_2_ also induced self-destruction of the vesicles, thus releasing quinone methide, which depleted cellular glutathione content and suppressed the antioxidant response of cancer cells. This synergistic effect of the nanoreactors resulted in a more efficient cancer cell killing and tumor ablation [18]. The herein reported results further support the validity of this pro-oxidant approach by demonstrating that P5 and P7, capable of producing a marked ROS-related cytotoxic effect, could represent new tools useful for counteracting cancer cell survival and the onset of chemoresistance.

## 2. Materials and Methods

### 2.1. Chemicals and Instruments

Monomer M5 (**5**) and copolymer P5 were prepared based on procedures recently reported [4]. Both the abovementioned synthesis, the experimental procedures performed for characterizing M5 and P5, and data results obtained are available in Appendix A. Data obtained by the physicochemical characterization of P5, which could be useful when making comparisons with the new copolymer (P7) prepared in this work, are also reported in tables available in Section 3.2, Section 3.3 and Section 3.4. Monomer M7 (**7**) was prepared on the base of opportunely modified procedures previously reported [19,20]. Experimental details are available in Section 2.2 of the present study. Melting points and boiling points are uncorrected. FTIR spectra were recorded as films or KBr pellets on a Perkin Elmer System 2000 instrument (PerkinElmer, Inc., Waltham, MA, USA). ^1^H and ^13^C NMR spectra were acquired on a Bruker DPX spectrometer (Bruker Italia S.r.l., Milan, Italy) at 300 and 75.5 MHz, respectively. Fully decoupled ^13^C NMR spectra are reported. Chemical shifts are reported in parts per million units relative to the internal standard tetramethylsilane (TMS = 0.00 ppm), and the splitting patterns are described as follows: s (singlet), d (doublet), t (triplet), q (quartet), m (multiplet), and br (broad signal). Mass spectra were obtained with a GC-MS Ion Trap Varian Saturn 2000 instrument (Varian, Inc., Palo Alto CA, USA; EI or CI mode; filament current: 10 mA) equipped with a DB-5MS (J&W) capillary column. Elemental analyses were performed with an EA1110 Elemental Analyser (Fison Instruments Ltd., Farnborough, Hampshire, UK).

HPLC analyses were performed on a Jasco model PU-980 instrument (JASCO Corporation, Hachioji, Tokyo, Japan), equipped with a Jasco Model UV-970/975 intelligent UV/Vis detector (JASCO Corporation, Hachioji, Tokyo, Japan) at room temperature. A constant flow rate (1 mL/min), UV detection at 254 nm, a 25 × 0.46 cm^2^ Hypersil ODS 5 mm column, and a mixture acetonitrile/water 6/4 as eluent were employed for the acquisitions. GC-FID analyses were performed on Perkin Elmer Autosystem (Varian, Inc., Palo Alto CA, USA) using a DB-5, 30 m, diameter 0.32 mm, film 1 mm capillary column. Column chromatography was performed on Merck silica gel (70–230 mesh). Dynamic Light Scattering (DLS) and Z-potential (ζ-p) determinations were performed on the same instrument and with the same modalities previously described [21]. Potentiometric titrations were performed with a Hanna Micro-processor Bench pH Meter (Hanna Instruments Italia srl, Ronchi di Villafranca Padovana, Padova, Italy). A thin layer chromatography (TLC) system employed aluminum-backed silica gel plates (Merck DC-Alufolien Kieselgel 60 F254, Merck, Washington, DC, USA), and detection of spots was made by UV light. The molecular weight of P7 was determined on a vapor pressure Knauer K-700 osmometer (Advanced Scientific Instruments Wissenschaftliche Gerätebau, Berlin, Germany) in MeOH at 45 °C.

All reagents and solvents were purchased from Merck (formerly Sigma-Aldrich, Darmstadt, Germany) and were purified by standard procedures. Commercial branched polyethyleneimine (PEI-*b*) (25 kDa) was purchased from Merck (formerly Sigma-Aldrich, Darmstadt, Germany); 2,2′-azobisisobutirronitrile (AIBN) was crystallized from methanol. Organic solutions were dried over anhydrous magnesium sulphate and were evaporated using a rotatory evaporator operating at a reduced pressure of about 10–20 mmHg.

### 2.2. 2-Methoxy-6-[(4-vinyl)benzyloxy]benzylamine Hydrochloride M7 (7)

#### 2.2.1. Chloromethylmethylether (**1**)

A mixture of methoxyacetic acid (22.50 g, 250.0 mmol) dissolved in freshly distilled thionyl chloride (27.3 mL, 376.0 mmol) was added with a drop of anhydrous dimethylformamide (DMF) and refluxed at 110 °C for 90 min in a single-necked flask with a CaCl_2_ valve. The stirring was then stopped, and the mixture was allowed to cool at r.t. and added with AlCl_3_ (0.35 g, 2.60 mmol), causing intense foaming. The new mixture was heated at 70 °C for 30 min. At the end, the reaction mixture was transferred into a Claisen instrument equipped with a deflammator arm, filled with glass spheres with an average diameter of 2 mm, and distilled at 1 a.t.m. by heating with a thermostated bath at 160 °C. Three fractions were collected as follows.

*Fraction 1.* Bp. = 53 °C, 1.05 g, which was discarded.*Fraction 2.* Bp. = 59–63 °C, 6.96 g, compound **1** polluted of the undecomposed intermediate acid chloride.*Fraction 3.* Bp. = 72–75 °C, 13.73 g, undecomposed intermediate acid chloride.

Fraction 3 was reacted again with AlCl_3_ (0.24 g, 1.80 mmol) as previously described and redistilled, collecting a single fraction (10.41 g) having a bp of 57–59 °C.

This fraction was joined to fraction 2 obtained from the first distillation, and the collected fractions were redistilled, obtaining the pure compound **1** (18.42 g, 229.0 mmol, 91% yield).

Bp 59–63 °C (Lit. [22] 59–60 °C). IR (film, ν cm^−1^) 1232, 1119 (C–O–C), 650 (C–Cl).

#### 2.2.2. 3-Methoxymethoxyanysole (**2**) 

A 60% mineral oil dispersion of NaH (4.6 g for actual 2.73 g, 114.0 mmol) was placed in a 250 mL, three-necked, round-bottomed flask equipped with a reflux condenser and a funnel, flushed with N_2_, and washed free of mineral oil with five small portions of pentane. DMF (90 mL) was added, and the suspension was stirred by a magnetic bar. The commercial 3-hydroxyanysole (13.98 g, 112.6 mmol) dissolved in 28 mL of dry DMF was added slowly (over 15 min) to the stirred mixture at 0 °C. The reaction was stirred at r.t. for 2 h. The chloromethyl methyl ether (9.97 g, 124.0 mmol) was added slowly, and the reaction was followed with TLC and was complete within 3 h after the addition of chloromethyl methyl ether. The reaction mixture was hydrolyzed with NaOH 10% (80 mL, pH = 14), and the aqueous layer was separated and extracted five times with ether (30 mL). The collected ether extracts were dried overnight on MgSO_4_. After removal of the solvent at reduced pressure, the crude oily product was purified by distillation under vacuum to yield compound **2** with a good degree of pureness (10.92 g, 64.9 mmol, 57.6% yield) [23]. Bp 111 °C (17 mm); purity 95% by HPLC, 94% by GC; IR (film, ν cm^−1^) 2836 (MeO), 1593, 1492 (C=C, phenyl), 1146, 1014 (C–O–C). ^1^H NMR (CCl_4_, 300 MHz, δ ppm) 3.44 (s, 3H), 3.74 (s, 3H), 5.13 (s, 2H), 6.2–7.5 (m, 4H). Anal. Calcd. for C_9_H_12_O_3_: C, 64.27; H, 7.19. Found: C, 64.57; H, 7.43. GC/MS: 168 (M^+^, 100%); 138 (C_8_H_10_O_2_^+^, 13%); 45 (C_2_H_5_O^+^, 36%).

#### 2.2.3. 2-Methoxy-6-methoxymethoxybenzaldehyde (**3**) 

3-methoxymethoxyanysole (16.33 g, 97.1 mmol) was dissolved in dry Et_2_O (180 mL), and the obtained solution was placed in a 500 mL, three-necked, round-bottomed flask equipped with a reflux condenser and a funnel and a freshly titrated 1.64 N butyllithium solution was added slowly (over 15 min). The mixture was stirred at r.t. for 90 min with formation of precipitate. Dry DMF (24.3 mL) dissolved in dry Et_2_O (36 mL) was added, and the suspension was stirred for additional 30 min and hydrolyzed with water (110 mL). The organic phase was separated, and the alkaline water was extracted four times with ether (30 mL). The collected ether extracts were dried overnight on MgSO_4_. After removal of the solvent at reduced pressure, the crude oily product was purified by column chromatography (eluent benzene/ethyl acetate 60/40) to achieve **3** (8.97 g, 45.6 mmol, 48% yield). IR (film, ν cm^−1^) 2839 (MeO), 1690 (C=O), 1597, 1475 (C=C, phenyl), 1156, 1076 (C–O–C) [19,20,24]. ^1^H NMR (CDCl_3_, 300 MHz, δ ppm) 3.51 (s, 3H), 3.90 (s, 3H), 5.25 (s, 2H), 6.66 (d, 1H, *J_o_* = 8.4 Hz), 6.79 (t, 1H, *J_o_* = 8.4 Hz), 7.46 (d, 1H, *J_o_* = 8.4 Hz), 10.51 (s, 1H). ^13^C NMR (75.5 MHz, δ ppm) 55.88, 56.31, 95.00, 105.00, 107.19, 115.44, 135.87, 160.00, 161.75, 189.13. GC/MS: 196 (M^+^, 27%); 150 (C_8_H_8_O_2_^+^, 61%); 45 (C_2_H_5_O^+^, 36%).

#### 2.2.4. 2-Methoxy-6-methoxymethoxybenzaldoxime (**4**)

A solution of **3** (1.08 g; 5.49 mmol) in 95% ethanol (13 mL) was treated with a solution of hydroxylamine hydrochloride (0.46 g, 6.59 mmol) in dry pyridine (2.2 mL) under stirring at room temperature for 1h 20 min and at 0 °C for 30 min to facilitate the oxime precipitation. The white solid was filtered, dried, weighed (0.82 g, 3.89 mmol). The mother liquors were concentrated to afford additional 0.0435 g of 2-methoxy-6-methoxymethoxybenzaldoxime (**4**) for an overall yield of 75%. Mp 131–135 °C; FTIR (KBr, ν cm^−1^) 3195 (OH), 2836 (MeO); 1627 (C=N), 1593, 1476 (C=C phenyl), 1072, 1000 (C–O–C).

#### 2.2.5. 2-Methoxy-6-methoxymethoxybenzylamine (**5**)

A solution of **4** (3.13 g; 14.8 mmol) in 95% ethanol (43 mL) was treated with an equal volume of 2 M NaOH followed by Raney nickel alloy (4.67 g) under stirring at room temperature for 90 min. The Raney nickel alloy was removed by filtration and washed with fresh ethanol. Filtrate and washings were combined, acidified with 0.8 M HCl (230 mL), and extracted with CH_2_Cl_2_ (30 mL). The aqueous phase was treated with solid KOH up to pH = 14 and extracted with diethyl ether (3 × 30 mL). The extracts after drying over anhydrous MgSO_4_ and removal of the solvent afforded 2-methoxy-6-methoxymethoxybenzylamine (**5**) (2.64 g, 13.40 mmol, 90% yield). Bp 70 °C/0.02 torr. Purity 98% by GC-FID. FTIR (KBr, ν cm^−1^) 3409, 3340 (NH), 2848 (MeO), 1594, 1472 (C=C phenyl), 1152, 1075 (C–O–C). ^1^H NMR (CDCl_3_, 300 MHz, δ ppm) 1.61 (bs, 2H); 3.48 (s, 3H); 3.82 (s, 3H); 3.89 (s, 2H); 5.20 (s, 2H); 6.58 (d, 1H, *J_o_* = 8.0 Hz); 6.73 (d, 1H, *J_o_* = 8.0 Hz); 7.14 (t, 1H, *J_o_* = 8.0 Hz).

#### 2.2.6. 2-Hydroxy-6-methoxymethoxybenzylamine Hydrochloride (**6**)

A mixture of **5** (2.93 g, 14.9 mmol), methanol (150 mL), and hydrochloric acid (5 mL) was heated at 62 °C under stirring for 20 min up to the disappearance of **5** (TLC, eluent benzene/ethyl acetate = 90/10). After removal of the solvent at reduced pressure, the solid residue was dissolved in the minimum amount of DMF and precipitated in chloroform to afford **6** in the form of pearly flakes (2.03 g, 10.71). The mother liquor after concentration and cooling afforded additional **6** (0.39 g, 2.04 mmol), overall yield 86%. Mp 211–214 °C. FTIR (KBr, ν cm^−1^) 3427 (OH) 3197, 3026 (NH_3_^+^), 2837 (MeO), 1603 (NH), 1573, 1503, 1473 (C=C phenyl), 1120 (C–O–C). ^1^H NMR (CD_3_OD, 300 MHz, δ ppm) 3.86 (s, 3H); 4.16 (s, 2H); 4.88 (s, 4H); 6.54 (m, 2H); 7.19 (m, 1H). ^13^C NMR (75.5 MHz, δ ppm) 33.66, 56.30, 103.12, 108.60, 109.21, 131.99, 158.21, 160.33.

#### 2.2.7. 2-Methoxy-6-[(4-vinyl)benzyloxy]benzylamine Hydrochloride M7 (**7**)

Sodium hydride (1.05 g, 25.3 mmol) as a 60% dispersion in mineral oil was washed three times with pentane under nitrogen and suspended in dry DMF (123 mL). The suspension was added with a solution of **6** (2.49 g, 13.11 mmol) in dry DMF (10.5 mL), stirred for 90 min, and treated with commercial chlomethylstyrene (1.9 mL, 15.11 mmol) under stirring at 40 °C for 3 h. The reaction mixture was hydrolyzed with 10% aqueous NaOH (40 mL, pH = 14) and extracted with peroxide-free diethyl ether (3 × 30 mL). The extracts after drying over anhydrous MgSO_4_ and removal of the last traces of DMF under vacuum afforded the free amine as crude oil (3.38 g, 12.00 mmol 95.8% yield). Without further purification, 3.24 g (12.01 mmol) of free oily amine was soon converted into its hydrochloride **7** by dissolution in dry diethyl ether (200 mL). The clear solution was cooled to 0 °C and treated under stirring up to saturation with dry gaseous hydrochloric acid (30 min). The hydrochloride **7**, precipitated as white flaky solid (3.61 g), was filtered, washed several times with fresh anhydrous ether, dried, and recrystallized from acetonitrile (ACN) to obtain a white crystalline solid (1.07 g, 3.50 mmol). The mother liquors placed in the refrigerator provided an additional 0.43 g of **7** for an overall yield of 43%. Mp 215–218 °C (acetonitrile). Purity 99% by HPLC. FTIR (KBr, ν cm^−1^) 3500–3000 (NH_3_^+^); 1599, 1474 (C=C, phenyl) 999 and 923 (CH_2_=CH). ^1^H NMR (CD_3_OD, 300 MHz, δ ppm) 3.90 (s, 3H), 4.20 (s, 2H); 5.16 (s, 2H); 5.23 (dd, 1H, *J_gem_* = 1.0 Hz; *J_cis_* = 10.9 Hz); 5.82 (dd, 1H, *J_gem_* = 1.0 Hz; *J_trans_* = 17.6 Hz); 6.70–6.80 (m, 3H); 7.33–7.45 (m, 5H). ^13^C NMR (75.5 MHz, δ ppm) 33.41, 56.45, 71.50, 105.06, 106.50, 110.26, 114.46, 127.43, 128.95, 132.46, 137.70, 137.73, 138.95, 159.05, 160.28. Anal. Calcd. for C_17_H_20_ClNO_2_: C, 66.77; H, 6.59;N, 4.58; Cl, 11.59. Found: C, 66.80; H, 6.58; N, 4.60; Cl, 11.62.

### 2.3. Preparation of Nanoparticulate Copolymer P7 by Radical Copolymerization in Solution

In a 25 mL tailed test tube equipped with a magnetic stirrer and carefully flamed under nitrogen, monomer M7 (**7**), DMAA, AIBN as radical initiator, and the freshly distilled anhydrous solvent were introduced in the ratios as reported in Section 3.2. The mixture thus obtained was subjected to three vacuum-nitrogen cycles to remove the oxygen. The clear solution was then siphoned into a 25 mL flask with screw cap and silicone septum. Nitrogen was then bubbled for 3 min in the solution, which was subsequently left under stirring at 60 °C for 72 h. The final yellow solution was evaporated at reduced pressure, achieving the crude copolymer, which was subjected to three cycles of dissolution in MeOH and precipitation in Et_2_O, obtaining P7 as white solid. P7 was subsequently subjected to fractioning as described in the following Section 2.3.1.

#### 2.3.1. Fractioning of P7

A solution of P7 in just enough MeOH was filtered and transferred in a three-neck round-bottomed flask equipped with a mechanic stirrer and a funnel. It was thermostated at 25 °C, and the clear solution (S1) was slowly added with Et_2_O until a milky precipitate (MP7-1) was obtained. MP7-1 was separated from the supernatant (S2) by centrifugation at 3000 rpm for 15′. S2 was treated as the starting solution (S1), obtaining a second milky precipitate (MP7-2). MP7-1 and MP7-2 were then dissolved in MeOH and precipitated in an excess of Et_2_O, obtaining the corresponding copolymers—namely, P7-High and P7-Low.

FTIR (KBr, ν cm^−1^) 3500 (NH_3_^+^); 2800–2900 (C–H stretching alkyl groups); 2000–1700 (aromatic overtones); 1649 (C=ONH); 1575 and 1510 (aromatic –C=C– stretching); 754 (*o*-disubstituted phenyl ring).

The unreacted monomer M7 (**7**) was recovered from the mixture of the combined solvents by evaporation at reduced pressure.

### 2.4. Average Molecular Mass (Mn) Determination of Copolymer P7

#### 2.4.1. Calibration Phase

Solutions of polyoxyethylene (PEO) with Mn 10,800 in MeOH were prepared at three different concentrations (c (mol/kg)) and were analyzed by the vapor pressure osmometer (VPO) technique at 45 °C. The quotients of measured values (MV) and the corresponding concentrations, i.e., MV/c (kg/mol), were determined. From these data (Section 3.3), by using the ordinary least squares (OLS) method, a linear regression curve was obtained, whose equation was Equation (1).
(1)y=73442x+500.92

Extrapolating it to concentration c = 0, *K_cal_* was determined and was found to be 501.

#### 2.4.2. Measurements Phase

Solutions of copolymer P7 in MeOH were prepared at three different concentrations c (g/kg) (Section 3.3) and were analyzed by VPO method at 45 °C. The ratios between the measurement values (MV) and concentrations (c) (kg/g) were plotted *vs.* concentrations (c), and by using the OLS method, a linear regression curve was obtained, whose extrapolation to concentration c = 0 provided the *K_meas_* (kg/g) value for P7 (Section 3.3). The average molecular mass (Mn) of P7 was determined with Equation (2) and is reported in Section 3.3.
(2)$$MW\;\left(\frac{g}{mol}\right)=\;\frac{K_{cal}}{K_{meas}}$$

### 2.5. Determination of NH_2_ Equivalents Contained in P7

The NH_2_ equivalents, in the form of hydrochloride salts contained in P7, were determined by volumetric titrations with a solution of HClO_4_ in acetic acid (AcOH) using quinaldine red as indicator [25]. Briefly, acetic anhydride (3 mL) was added to a solution of HClO_4_ 70% (1.4 mL) in AcOH (80 mL), obtaining a colorless solution that was left under stirring at room temperature overnight. The clear yellow solution was made up to 100 mL with AcOH and standardized with potassium acid phthalate. The title of solution was found to be 0.1612 N. A sample of P7 (350.1 mg) was dissolved in AcOH (5 mL), treated with 2 mL of a solution of mercury acetate (1.5 g) in AcOH (25 mL), added with a few drops of a solution of quinaldine red (100 mg) in AcOH (25 mL), and titrated with the standardized solution of HClO_4_ in AcOH using a calibrated burette with needle valve (0.02 mL). The very sharp end points were detected by observing the disappearance of the red color. Standardization and titrations were made in triplicate, and the results are reported as means ± standard deviation (SD). The content of NH_2_ was expressed both as µequiv. of NH_2_ per µmol of P7 and as µequiv. of NH_2_ per gram of P7 and reported in Section 3.4.

### 2.6. Potentiometric Titration of P5 and P7

Potentiometric titrations were performed at room temperature to construct the titration curves of P5 and P7. The copolymers (20–30 mg) were dissolved in 30 mL of Milli-Q water (mQ), then were treated with a standard 0.1 N NaOH aqueous solution (1.5 mL, pH = 9.34 (P5) and 9.54 (P7)). The solutions were potentiometrically titrated by adding 0.2 mL aliquots of a standard 0.1 N HCl aqueous solution, up to total 3.0 mL, and measuring the corresponding pH values [26]. For comparison purposes, commercial branched polyethyleneimine (PEI-*b*) 25 kD was titrated under the same conditions. Titrations were made in triplicate, and the determinations are reported as mean ± SD.

### 2.7. Z-Potential (ζ-p) and Dynamic Light Scattering (DLS) Analysis of P7

ζ-p and dynamic light scattering (DLS) measurements of P7 were performed in aqueous solutions (mQ) (pH = 7.4) at a fixed concentration of 1 mg/mL and at the physiological temperature of 37 °C. The excitation light source was a 4 mW He–Ne laser at 633 nm, and the intensity of the scattering angle was fixed at 173 °C. Results were the combination of three 10-min runs for a total accumulation correlation function (ACF) time of 30 min. From the relaxation time determined, the apparent diffusion coefficient, D, was estimated at the actual co-polymer concentration. The apparent hydrodynamic radius (R_hyd_) of the particles was calculated through the Stokes–Einstein equation [27] and is reported as the mean of three measurements ± SD.

PDI value is reported as the mean of three measurements ± SD made by the instrument on the sample. The ζ–p was measured at 37 °C in mQ water as a medium, and an applied voltage of 100 mV was used. The P7 sample was loaded into pre-rinsed folded capillary cells, and twelve measurements were performed.

### 2.8. In Vitro Evaluation of Cytotoxicity of P5 and P7 against Human Neuroblastoma (NB) Cells

#### 2.8.1. Cell Culture Conditions and Treatments

HTLA-230 human NB cells were kindly provided by Dr. L. Raffaghello (G. Gaslini Institute, Genoa, Italy). ETO-resistant HTLA (HTLA-ER) were selected by HTLA-230 parental cells as previously reported [28]. To determine the cytotoxic effects of P5 and P7, experiments were carried out by treating both cell populations for 24 h with increasing concentrations of P5 and P7 (0.1–50 μM). In addition, for comparative purposes, increasing concentrations of M5 and M7 (5–250 µM) were also tested in the same conditions.

Stock solutions of P5/P7 and M5/M7 were prepared in DMSO, employing final amounts of solvent unable to affect cell responses.

#### 2.8.2. Cell Viability Assay

Cell viability was determined as previously described [21].

#### 2.8.3. Detection of Reactive Oxygen Species (ROS) Production

The production of ROS was evaluated by using 2′-7′ dichlorofluorescein-diacetate (DCFH-DA; Sigma) [29] as previously reported [21].

### 2.9. Statistical Analyses

Data are expressed as means ± SEM from at least four independent experiments. Statistical significance of parametric differences was determined by one-way analysis of variance (ANOVA) and Dunnett’s test for multiple comparisons.

## 3. Results

### 3.1. Synthesis and Spectrophotometric Characterization of 2-Methoxy-6-[(4-vinyl)benzyloxy]benzylamine Hydrochloride M7 (**7**)

The multistep synthesis of M7 (**7**) was performed according to Scheme 2 and following modified versions of previously reported procedures [19,20,22,23,24].

Although chloromethylmethylether (**1**) is commercially available, due to its high cost, it was prepared starting from methoxyacetic acid and following one of the two procedures previously reported [22] and shown in the first step of Scheme 2. This procedure was preferred because it did not need to separate the intermediate acylic derivative and allowed **1** to be achieved with a good degree of purity and a higher yield. Note that to have good reaction outcomes, SOCl_2_ needed to be freshly distilled and AlCl_3_ to be anhydrous. Compound **1** was used to transform the commercial 3-hydroxyanisole in compound **2** according to the procedure reported by Winkle and Ronald [23]. The obtained oily product was analyzed by HPLC and GC, demonstrating a good purity (95% and 94%, respectively), and by FTIR and NMR spectroscopy, providing data in accordance with literature. Additionally, the elemental analysis confirmed is structure. The reaction performed to obtain the aldehyde **3** was based on the ortho-directed metallation of 3-methoxymethoxyanisole (**2**) [24], for which we formerly highlighted some very interesting results [20]—different from those reported previously [24]. The aldehyde **3**, was transformed into its benzaldoxime **4**, and its reduction afforded the benzylamine **5** as an easily distillable liquid. The deprotection of **5** afforded the phenolic hydrochloride **6** in good yield. Unfortunately, after the last two steps—i.e., the alkylation of **6** with 4-chloromethylstyrene (4-CMSTY) to produce the crude free base and then the monomer M7 (**7**)—an overall of only 26% yield from **3** was achieved. Consequently, attempts to follow an alternative route were performed. In particular, we tried to transform **3** into the 2-methoxy-6-hydroxybenzaldehyde and to alkylate it with 4-CMSTY in order to obtain the alkylated aldehyde styrene derivative, but the subsequent conversion of the aldehyde group into the amine hydrochloride necessary for achieving our goal highlighted additional difficulties, and consequently, this alternative path was no longer investigated. Even if the yield of M7 was limited, the degree of purity was excellent (99% HPLC), and the FTIR and NMR spectra confirmed its structure.

### 3.2. Preparation of Copolymer P7 by Radical Copolymerizations in Solution and Spectroscopic Characterizations

Recently, by performing the simple and low-cost radical copolymerization in solution of M5, a copolymer P5 was obtained, characterized by excellent physicochemical properties [4]. Therefore, also in this study, before resorting to more sophisticated polymerization procedures, such as atom transfer radical polymerization (ATRP) and reversible addition-fragmentation chain transfer (RAFT), we performed preliminary studies of radical polymerization in solution of M7. Indeed, note that, even if ATRP and RAFT allowed increased control of the polymers’ molecular weight, molecular architecture, and composition while maintaining a low polydispersity, both ATRP and RAFT presented significant drawbacks, which we preferred to avoid. Concerning ATRP, high concentrations of catalyst, which standardly consist of a copper halide and an amine-based ligand, are required for the reaction. The removal of the copper from the final polymer is often tedious and expensive [30]. In addition, ATRP is also a traditionally air-sensitive reaction normally requiring freeze-pump thaw cycles [31]. As for RAFT, unstable over long time periods, highly colored and sulfur-containing RAFT agents are required, and most of them are only suitable for a limited set of monomers. The synthesis of a RAFT agent typically requires a multistep synthetic procedure and subsequent purification [32]. RAFT agents can have a pungent odor due to gradual decomposition, yielding small sulfur compounds. Finally, in sight of a future biomedical application of a polymer, the possible presence of sulfur and color due to the RAFT procedure may be undesirable. In the present case, as already observed for M5 [4], the preliminary studies of radical polymerization in solution had showed that M7 omopolymerized and copolymerized easily with different comonomers, including DMAA, which is considered an optimal hydrophilic comonomer for preparing polar supports [19,33]. The couples solvent/initiator used included water/ammoniumpersulfate (APS) and methanol or DMF with AIBN, which allowed conversions in the range 20–94%. Consequently, in this work, DMAA was selected as comonomer, and copolymerization of **7** was performed in MeOH/AIBN at 60 °C, achieving a conversion of 85% (Scheme 3).

The experimental data of the copolymerization of M7 are reported in Table 1, together with data previously reported for the copolymerization of M5 to obtain P5 [4].

As shown in Table 1, P7 was obtained with higher percentage of conversion than M5, probably due to the longer reaction time. The copolymer P7 was purified by repeated cycles of dissolution/precipitation using MeOH as solvent and Et_2_O as non-solvent. In addition, a sample of P7 was subjected to fractioning. Fractioning was performed at 25 °C, adding an excess of Et_2_O to the subsequently isolated milky precipitates of copolymers, achieving high Mn and low Mn copolymers (P7), free from traces of monomer M7 and co-monomer (DMAA), as confirmed by the NMR spectrum, lacking the typical signals of the three double doublets, each one integrable for 1H, of the vinyl system. In the FTIR spectrum of P7 (Figure 1), the presence of bands deriving both from M7 and DMAA were observable. In particular, an intense broad absorption around 3000–3500 cm^−1^ due to the NH_3_^+^ groups of M7 (out of scale), overtones of its phenyl rings under 2000 cm^−1^, and two sparkly bands at 1510 and 1575 cm^−1^ indicating the C=C bonds were observable. In addition, the amide band (C=ONH) of the comonomer DMAA was also detectable around 1650 cm^−1^.

Similar to P5, the copolymer P7 was soluble in water, methanol, DMSO, and DMF, while insoluble in petroleum ether, diethyl ether, and acetone.

### 3.3. Average Molecular Mass (Mn) Determination of Copolymer P7

#### 3.3.1. The Technique

The average molecular mass (Mn) of P7 was determined by using the vapor-pressure osmometry (VPO) method in MeOH at 45 °C. VPO is an experimental technique for the determination of a polymer’s number average molecular weight, Mn. It works by taking advantage of the decrease in vapor pressure that occurs when solutes are added to pure solvent. This technique can be used for polymers with a molecular weight of up to 20,000, though accuracy is best for those below 10,000 [34].

Higher polymers can be analyzed using other techniques, such as membrane osmometry and light scattering. Since 2008, VPO has faced competition from matrix-assisted laser desorption ionization mass spectrometry (MALDI-MS), but VPO still has some advantages when fragmentation of samples for mass spectrometry may be problematic [35].

A typical vapor phase osmometer consists of two thermistors, one with a polymer-solvent solution droplet adhered to it and another with a pure solvent droplet adhered to it. A number of syringes are employed to provide thermistors with pure solvent or solution drops. The thermistors are located in a cell where the gas phase is saturated with solvent vapor. The cell temperature is electronically controlled and maintained with an accuracy of ±1 × 10^−3^ °C. The operating temperature can be selected in the range between 20 and 130 °C. An electric circuit measures the bridge output imbalance difference between the two thermistors. The increasing vapor pressure of the solution droplet leads to an increase of temperature. Once equilibrium is reached, a constant measurement value is achieved. This particular ∆*T* between the thermistors is always proportional to the number of particles or number of moles dissolved in the solution. Consequently, when the sample concentration is known, average molecular mass (Mn) can be determined.

#### 3.3.2. Calibration

By using PEO 10,800 as the selected standard for our determinations, in the first phase of calibration, a linear regression curve was developed using the OLS method, whose data are reported in Table 2, and which allowed calculation of K_cal_, corresponding to the value on the *y* axis when c (mol/kg) was zero. In the second phase, measurements were executed on methanol solutions of P7 whose concentrations (g/kg) are reported in Table 2, and the K_meas_ (kg/g), corresponding to the value on the *y* axis when c (g/kg) is zero, was provided by the instrument (Table 2). K_meas_ was used to estimate the Mn of P7 according to Equation (2), reported in Section 2.4.2.

### 3.4. Determinations of NH_2_ Equivalents Contained in P7

To determine the NH_2_ equivalents contained in P7 and have evidence of its content of protonated groups, the titration of amine hydrochlorides with HClO_4_ solution in AcOH in the presence of mercuric acetate and quinaldine red as indicator [25] proved to be simple and affordable. The method is cheap, fast, and its accuracy was secured by a sharp endpoint of titration, while its reliability was demonstrated by the reproducibility of results (Table 3). For comparison purposes, Table 3 also collects the results obtained previously for copolymer P5 [4].

### 3.5. Particle Size, ζ-p and PDI of P7

The apparent hydrodynamic radius (R_hyd_) of P7 was determined by DLS analysis, and ζ-p measurements were carried out to determine its surface charge. The results are reported in Table 3, where the results previously obtained for P5 are also included for comparison purposes.

P7 particles showed an average size of 220 nm, which denotes a lower size with respect to the particle size of P5 [4]. The absence of the four carbon atoms alkyl chain, as linker between the cationic group and the aromatic rig, probably led to a more impacted macromolecular structure, consequently characterized by particles with reduced dimensions. The water solutions of P7 proved to remain clear over time, both at room temperature and under heating, assuring a good stability in solution, as confirmed also by the value of ζ-p (+50 mV), which was significantly higher than the value of 30 mV, which is considered a threshold value below which there can be low stability in solution and a tendency to form aggregates.

As expected, ζ-p of P7 was positive, confirming its cationic character, even if the value was slightly lower than that measured for P5 [4], thus establishing a minor cationic asset. Rationally, the presence of the two ether groups in *orto*-position to the cationic function created a clutter or shielded the charge itself and consequently reduced its positivity. Polymer particles with high positive ζ-p are usually capable of faster absorption on cells’ surface by electrostatic interactions, with an eventual easier internalization than particles with lower positive ζ-p; but on the other hand, highly positive particles could result in strong cytotoxicity. A softened cationic character could allow P7 to maintain activity as a membrane disruptor, with reduced toxicity for mammalian cells.

Furthermore, ζ-p of P7 was in accordance with those of cationic polystyrene-based nanoparticles recently prepared by co-polymerizing styrene with the *N*-(2-(methacryloyloxy)ethyl)-*N*,*N*-dimethyltetradecane-1-ammonium bromide (MDTP) as active monomer [36]. A compound named CNPS-3, with 40% content of cationic monomer but a lower Mn compared to P7, displayed an identical value of ζ-p (+50 mV).

### 3.6. Potentiometric Titration of P5 and P7

The herein tested copolymers P5 and P7 had neither quaternary nor permanently protonated ammonium groups but were both characterized as having reversibly protonable primary amine groups, depending on the pH value of the environment. The protonation of amine groups of nanomaterials, intended to work as an antibacterial or chemotherapeutic agent, is essential for having a positively charged surface, which is crucial for interacting with the surface of cells and for providing significant cytotoxic activity. Therefore, to suggest a possible clinical application of P5 and P7, it was important to know the pH values at which they can be protonated and, mainly, if they would be protonated in the physiological pH range of 4.5–7.5. To obtain this information, the potentiometric titrations of P5 and P7 was carried out according to Benns et al. [26], and the data are reported in Table 4.

The titration curves of P5 and P7 were obtained by graphing the measured pH values *vs*. the aliquots of HCl 0.1N added (Figure 2a). Subsequently, from titration data, the dpH/dV values were computed and are reported in Table 2. By reporting in the graph these values vs. those of the corresponding volumes of HCl 0.1N, the first derivative lines of the titration curves were obtained (Figure 2b). The maxima of these latter curves corresponded to the volumes of HCl necessary to have P5 and P7 in the protonated forms. Interestingly, for both copolymers two maxima were observed, thus establishing the existence of a two-step protonation process. Data reported in the last three rows of Table 4 proved that both P5 and P7 in the physiological pH range existing outside the cells should be completely protonated, as desired, thus favoring electrostatic interactions with negatively charged cells surface.

Furthermore, as extensively reported, cationic materials, such as those used to bind and transport genetic material inside the cell for gene-therapy purposes, were found to be internalized also by endocytosis [25,37]. Consequently, once inside the cells and within the endosome, cationic materials need to escape it so as to not be degraded into the lysosomes. Several studies have reported that cationic gene delivery systems achieve this goal by acting as a “proton sponge” able to attract protons inside the endosome, and consequently causing osmotic swelling and bursting [25,37]. In this regard, the proton sponge activity of cationic macromolecules mainly depends on their buffer capacity (β = dc (HCl)/d(pH)) [38] (defined as the percentage of amino groups becoming protonated from pH 7.4 to 5.1) and on their average buffer capacity (βave = dV(HCl)/dpH(1)) [39] in the pH range 4.5–7.5, which improves with the increasing of the values of β and βave. Thus, to predict the capability of P5 and P7 to avoid premature degradation and inactivation, evaluating their β and βave was essential. The potentiometric titration data were exploited to compute β and βave according to the abovementioned formulas in brackets (pH range 4.5–7.5). The max β values observed for P5 and P7 and their values of βave are reported in Table 5 and compared with those obtained for commercial branched PEI-*b* (25 kDa), a reference standard recognized for having good buffer capacity, which was assayed in the same titration conditions.

By reporting in graph all the β values determined for P5, P7, and PEI-*b* in the desired pH range vs. the values of corresponding pH, their buffer capacity curves were obtained, which clearly showed the max values reported in Table 5 (Figure 3a,b).

As shown in Figure 3a, PEI-*b* had more than one maximum of β, and in the pH range of interest, two maxima were detectable, particularly at pH values of 6.81 and 7.33. However, when PEI-*b* results were reported in a graph also containing those of P5 and P7, the maxima of PEI-*b* were no longer visible because the maxima were extremely lower than those of P5 and P7, thus causing the flattening of the curve observed Figure 3a to a flat line (Figure 3b). These findings established that, at fixed pH points within the pH range of interest, both P5 and P7 possessed a buffer capacity far higher than that of PEI-*b*. Both P5 and P7 showed max values of β at pH significantly different from those of PEI-*b*; P5 showed a single βmax 2-fold lower than the βmax shown by P7 at similar pH, while P7 showed an additional very intense βmax at lower pH. Interestingly, when βave values were considered (Table 5), P5 and P7 had lower values than that of PEI-*b*, as is also observable in Figure 4.

### 3.7. Cytotoxic Effect of P5 and P7 on Two Human NB Cell Lines

To evaluate the potential of P5 and P7 as new promising therapeutic devices, two NB cell lines, one sensitive (HTLA-230) and the other resistant to ETO (HTLA-ER), were exposed to both compounds administered in a concentration range between 0.1–10 µM for 24 h. For additional comparison, the cytotoxic effects of M5 and M7, the monomers from which P5 and P7 were obtained, were tested on NB cells treated with increasing doses (5–250 µM) of them. As reported in Table 6, M5 and M7 reduced by 50% (LD_50_) the viability of both cell populations at the concentration of 250 μM and 100 µM, respectively, thus establishing a minimal cytotoxic activity, although significantly more marked in M7.

#### 3.7.1. Dose-Dependent Effects of P5 on NB Cell Viability

The treatments with P5 significantly reduced cell the viability of both HTLA-230 (Figure 5a, HTLA) and HTLA-ER cells (Figure 5b, ER) in a concentration-dependent way, starting from 1 µM.

By converting the bar graphs to dispersion graphs, the curves showing the cell viability decrement as a function of P5 concentrations were obtained. This analysis showed that the decrease in cell viability was rather linear up to the concentration of 2.5 μM for ER cells and of 5 μM for the HTLA cells, while a plateau in cell viability was reached for concentrations major of these values (Figure 6). From the Equations (3) and (4) associated to the linear regressions achievable form these tracts of curve by the Ordinary Least Squares (OLS) method (Figure 6b), it was possible to determine the LD_50_ values for P5. The values of LD_50_ were calculated both for HTLA and ER cells, respectively (Table 6), thus establishing a far higher toxicity of P5 on ER than on HTLA cells.
(3)y=−11.153x+97.532
(4)y=−23.703x+102.2

As inferred by the dispersion graphs and the associated linear regression models, different from M5, the effects of P5 on the two cell populations were dissimilar, and interestingly, P5 displayed more toxicity on ER than on HTLA cells, determining the same effects at a mid-dose. This different sensitivity of the two cell populations probably is related to the dissimilar capability of the P5 to interact selectively with the cell membrane. In fact, the data could lead one to hypothesize that during the acquisition of resistance, the membrane of ER cells changes, becoming more similar to that of bacteria. However, this is a mere hypothesis that needs further investigation.

#### 3.7.2. Dose-Dependent Effects of P7 on NB Cell Viability

The results obtained in NB cells treated with P7 are reported in Figure 7. Similar to P5, the effects of P7 on the two cell populations were dissimilar, and even if cytotoxic concentrations were lower than those of P5, P7 also displayed more toxicity on HTLA-ER (Figure 7b) as compared with HTLA-230 cells (Figure 7a).

The bar graphs were converted to dispersion graphs (Figure 8a), and the curves showing the cell viability decrement as a function of P7 concentrations were obtained. This analysis showed that the decrease in cell viability was linear (*R*^2^ = 0.9875 and 0.9900 for HTLA-230 and EHTLA-R, respectively) up to the concentration of 5 μM for both cells’ populations. From the Equations (5) and (6) associated to the linear regressions achievable form these tracts of curve, by the Ordinary Least Squares (OLS) method (Figure 8b), it was possible to determine the LD_50_ values for P7. The values of LD_50_ were calculated both for HTLA and ER cells, respectively (Table 6), thus confirming higher toxicity of P7 on ER than on HTLA cells.
(5)y=−9.2103x+96.679
(6)y=−12.017x+98.808

Collectively, the values of LD_50_ obtained for P7 were slightly higher than those shown by P5, toward both HTLA and ER cells, thus demonstrating a higher cytotoxic potency of P5. Indeed, according to other studies, macromolecules with lower molecular weight, as was P5 if compared with P7, are more cytotoxic than macromolecules with higher molecular weight [40]. In addition, considering that P5 and P7 had an LD_50_ between 2 μM and 4 μM—while the cationic peptide lactoferricin B reported by Eliassen [2] affected the viability of both MYCN-amplified and not amplified NB cells with a LD_50_ values in the range 15.5–60 µM—it was conceivable that both cationic copolymers herein reported could be a very much more efficacious strategy with which to counteract NB progression and chemioresistance than the peptide previously reported [2].

### 3.8. Dose Dependent ROS Production in HTLA and ER Cells

Since several chemotherapeutic drugs, including ETO, exert anticancer effect by increasing ROS production [1], and HTLA-ER cells display higher levels of antioxidants as compared with HTLA-230 [28], the effects of P5 and P7 on ROS generation were investigated.

Remarkable differences in the pro-oxidant effects of P5 and P7 were found, thus confirming the cytotoxicity data. In particular, P5 was more effective than P7, determining a significant increase in ROS levels at mid-dose (2.5 µM (P5) versus 5 µM (P7)), both in HTLA-230 and in HTLA-ER cells (Figure 9 and Figure 10).

Moreover, as observed in Figure 9, P5 was more effective on chemoresistant cells, inducing a 116% increase of ROS production in HTLA-ER as compared with a 65% increase detected in HTLA-230 cells. Differently, P7 induced a comparable effect on ROS generation in HTLA-230 and HTLA-ER cells according to the cell viability data (Figure 10).

As reported in the literature [2], among the established events responsible for the cytotoxic effect exerted by cationic molecules, a collapse of the outer and inner mitochondrial membranes, which could exacerbate ROS generation, was included. Indeed, our results establish that both copolymers were effective in enhancing ROS production in both NB cells.

The more relevant pro-oxidant effect evoked by P5, as compared with P7, could be due to the lower average molecular mass of P5, which makes for its easier and faster entry into the cells, also through small pores. Furthermore, the chemical structure of P7, derived from a monomer possessing two aromatic rings and a cationic group separated from the benzene ring by a single methylene, confers rigidity to the macromolecule and steric hindrance on the protonated nitrogen atoms, with possible hampering of the ionic interactions necessary for the activity.

By comparing cell viability with ROS production, a direct linear correlation between ROS production and the cytotoxicity promoted by two polymers in both NB cell lines was found (Graph S1).

Although more in-depth investigations are currently underway to clarify the mechanism of action of P5 and P7, on the basis of previous studies [2,16], we are confident hypothesizing that the cytotoxic effect of both copolymers could be due to their ability to interact with the cell membrane, thus causing direct destabilization and could also be due to their ability to enhance permeabilization through the formation of pores. Subsequently, P5 and P7 could be internalized into the cytoplasm either through the pores or by endocytosis, as reported for cationic gene delivery systems. In this latter case, due to their high buffer capacity, P5 and P7 could escape the endosome and lysosome degradative attack. Subsequently, they could act on mitochondria by stimulating ROS production, thus triggering cell death. Furthermore, it is conceivable that the greater susceptibility of HTLA-ER cells, with respect to HTLA-230, could be due to changes in the constituents and charge of the membrane, probably occurring when resistance was developed, which, becoming more negative and fluid, could allow easier interactions with cationic polymers and easier entry of the polymers into the cells. Collectively, although ROS are essential for provoking the cytotoxic effect both on HTLA-230 and HTLA-ER cells, crucial to inducing ROS production is the entry of copolymers into the NB cells, which can only happen after the copolymers’ electrostatic interaction with the cells’ cytoplasmic membrane. However, further investigations are necessary to experimentally demonstrate those assumptions. In this regard, due to the abovementioned membrane modifications, which probably occurred in becoming resistant to ETO, electrostatic interaction with HTLA-ER cells could have been easier, thus determining a higher cytotoxic effect on the NB cells, regardless of their chemoresistance.

## 4. Conclusions

This study established that the hydrophilic cationic random copolymer P5, already reported to be endowed with a potent antibacterial activity against multi-drug resistant superbugs belonging both to Gram-positive and Gram-negative species, and the newly prepared copolymer P7 possess significant ROS-related cytotoxic effects against HTLA-230 and HTLA-ER NB cells. P5 and P7 were prepared by simple and low cost radical polymerization in solution, without resorting to more sophisticated and expensive procedures such as ATRP and RAFT. The nanoscale dimensions of the particles; their water-solubility; their positively charged surface, essential for exerting cytotoxic activity; the presence of nitrogen atoms, which proved to remain protonated in the whole physiological pH range; and their stability in solution and excellent buffer capacity were the main physicochemical characteristics of P5 and P7, establishing their suitability for future biomedical applications. Interestingly, both P5 and P7 displayed a stronger cytotoxic activity on HTLA-ER cells, thus being promising template macromolecules for the development of new chemotherapeutic agents able to fight NB chemoresistance.

Although additional studies concerning the possible mechanisms of action of P5 and P7 are necessary, a rational hypothesis, supported by previously reported investigations and by the herein performed experiments, has been provided. Moreover, since cationic devices similar to P5 and P7 were not cytotoxic to healthy eukaryotic cells, the herein described devices could represent new well-performing alternative compounds able to selectively counteract cancer growth and chemoresistance without inducing secondary drug-related toxic effects.

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
