# Peer review of "Synthesis of Polystyrene-Based Cationic Nanomaterials with Pro-Oxidant Cytotoxic Activity on Etoposide-Resistant Neuroblastoma Cells"

_nanomaterials, 2021, doi:10.3390/nano11040977_

Round 1

Reviewer 1 Report

The manuscript titled “Synthesis of Polystyrene-Based Cationic Nanomaterials with ROS-related Cytotoxic Activity on Etoposide-Resistant Neuro-3 blastoma Cells” by authors demonstrated that cationic polymer could induce oxidative stress (oxidation therapy) to overcome drug-resistence for cancer treatment. Due to the drug resistance of most cancers to chemotherapeutic drugs, chemotherapy is difficult to completely eliminate tumors. The oxidation therapy provides the possibility to solve the problem of drug resistance. In general, this work seems to be interesting and I do recommend the acceptance of this work provided that authors can well address the following questions.

  1. The method to prepare the nanoparticle should be provided to fully understand this study.
  2. How is the PDI of P5 and P7 nanoparticle? It is possible to provide the TEM images of nanoparticle?
  3. Is it possible to prepare the polymer using controlled radical polymerization method, such as RAFT (reversible addition–fragmentation chain transfer) and ATRP (atom transfer radical  polymerization)? It is important to control the quality of polymer.
  4. Why P5 and P7 showed higher toxicity on ETO-resistant cell than ETO-sensitive cell? It is difficult to understand. Because the cytotoxic mechanism of ETO is also ROS. 
  5. ROS in title should be given in full name.
  6. A control on cytotoxicity for normal cell should provide to demonstrate the selectivity.
  7. The authors should cover the new progress of oxidation therapy in introduction for research background, but not limited to cationic polymer (e.g., Angewandte Chemie International Edition, 2020, 59(32): 13526-13530; Angewandte Chemie International Edition, 2017, 56(45): 14025-14030).
  8. Some small mistakes, for example, "For comparison also the cytotoxic effects of M5 and M7, the monomers from which 577 P5 and P7 were obtained, were tested on NB cells treated with increasing doses (5-250 578 µM) of them. As reported in Table 4, M5 and M7 reduced by 50 % (LD50) the viability of 579 both cell populations at the concentration of 250 μM and 100 µM respectively" here Table 4 should be Table 6. "The treatments with P5 reduced cell viability of both HTLA-230 (Figure 5a, HTLA) 583 and HTLA-ER (Figure 5b, ER) in a concentration-dependent way starting from 1 µM." 1 µM should be 0.1 µM.

Author Response

The manuscript titled “Synthesis of Polystyrene-Based Cationic Nanomaterials with ROS-related Cytotoxic Activity on Etoposide-Resistant Neuroblastoma Cells” by authors demonstrated that cationic polymer could induce oxidative stress (oxidation therapy) to overcome drug-resistence for cancer treatment. Due to the drug resistance of most cancers to chemotherapeutic drugs, chemotherapy is difficult to completely eliminate tumors. The oxidation therapy provides the possibility to solve the problem of drug resistance. In general, this work seems to be interesting and I do recommend the acceptance of this work provided that authors can well address the following questions.

We thank the Reviewer for the positive general comments on our study and we hope to further satisfy him/her with our following responses to his/her requests.

  • The method to prepare the nanoparticle should be provided to fully understand this study.

We kindly point out to the Reviewer that the method for preparing the nanoparticulate copolymers P5 and P7, has already been reported in the original non-revised version of this work.

As regards P5, since the method has already been described in a previous and recent manuscript (Ref 4), the procedure has been included in the Supporting Information (SI) (Section S2.1), while as regards P7, the procedure is described in the Section 2.3 (lines 275-298).

However, for greater clarity, and to highlight that when we talk about nanoparticles, we refer precisely to the copolymers P5 and P7, the title of Section S2 of the SI and the title of Section 3.2 (line 275) of the main text have been modified as follows:

Section S2. Preparation and Characterization of Nanoparticulate Copolymer P5 [1].

2.3. Preparation of Nanoparticulate Copolymer P7 by Radical Copolymerization in Solution

2)  How is the PDI of P5 and P7 nanoparticle? It is possible to provide the TEM images of nanoparticle?

We are forced again to point out to the Reviewer that PDI of P5 and P7 have already been reported in the original version of this work, and precisely in Table 3. In this regard, Table 3 also contains the hydrodynamic size and the values of Z-potential of the particles of P5 and P7. Since this collection of essential information concerning P5 and P7 particles is already present, and it was obtained by performing a recognized as valid method, by carrying out DLS analyses, we find useless and redundant performing additional analyses to obtain the same information. On the other hand, we do not hide to the Reviewer, that being a small academic laboratory, despite being very well equipped in terms of instruments, we cannot afford to have more equipment dedicated providing the same information. We therefore kindly ask the Reviewer to be satisfied with the data already provided through the instrument in our possession.  

However, to focus better the readers’ attention to these data (Z-ave (nm), Z-potential and PDI) present in Table 3, we have moved the Section dedicated to the relative discussion closer to the Table. Please see lines 511-538.

  • Is it possible to prepare the polymer using controlled radical polymerization method, such as RAFT (reversible addition–fragmentation chain transfer) and ATRP (atom transfer radical polymerization)? It is important to control the quality of polymer.

Of course. But, as we have reported in the non-revised version of the main text, in preliminary studies of radical polymerization in solution, M5 and M7 had shown to omopolymerize and copolymerize easily with different comonomers including DMAA, providing from good to high percentages of conversion. So, since the method used by us is very simple, reliable and does not require special operative and purification procedures and/or particular, expensive, and/or problematic reagents, we thought not necessary to recover to more sophisticated and expensive polymerization methods, as RAFT or ATRP. Moreover, we were not interested in highly controlled polymerizations which provide copolymers with increased control of molecular weight, molecular architecture and composition while maintaining a low polydispersity, but in obtaining random copolymers, easily achievable also without performing RAFT or ATRP. We agree with the Reviewer that to control the quality of polymer is important, but from the experience acquired with P5, and as the several characterization data confirmed, the quality of P5 was very good and above all, concerning its bioactivity, P5 proved to be an excellent antibacterial device. In our opinion, and as the Reviewer himself has recognized, in the present study, both P5 and P7 have demonstrated to perform very well as anticancer devices, thus establishing that their polymeric structure obtained with the method of radical polymerization in solution is the winning one. Furthermore, both ATRP and RAFT methods present significant drawbacks, which we preferred to avoid. Concerning ATRP, high concentrations of catalyst, which standardly consists of a copper halide and an amine-based ligand, are required for the reaction. The removal of the copper from the final polymer is often tedious and expensive. In addition, ATRP is also a traditionally air-sensitive reaction normally requiring freeze-pump thaw cycles. As for RAFT, unstable over long time periods, highly colored, and sulfur containing RAFT agents are required, and most of them are only suitable for a limited set of monomers. The synthesis of a RAFT agent typically requires a multistep synthetic procedure and subsequent purification]. RAFT agents can have a pungent odor due to gradual decomposition to yield small sulfur compounds. Finally, in sight of a future biomedical application of a polymer, the possible presence of sulfur and color due to the RAFT procedure may be undesirable. Anyway, to clarify better our synthetic choices, a discussion on the question and on the advantages and disadvantages of using ATRP and RAFT was added in the revised version of the manuscript (Section 3.2, lines 409-435).   

  • Why P5 and P7 showed higher toxicity on ETO-resistant cell than ETO-sensitive cell? It is difficult to understand. Because the cytotoxic mechanism of ETO is also ROS.

We thank the Reviewer for having noted this interesting and unexpected but welcomed outcome of our research and this property of our polymers. Since to counteract the resistance of malignant cells to conventional chemotherapeutics is the goal constantly pursued, the more cytotoxicity of P5 and P7 on NB cells resistant to ETO, confirms the relevance of our research.

As we have already highlighted in the main text at the end of Section 3.8, more in-depth investigations are currently underway to clarify the mechanism of action of P5 and P7. However, based on previous studies [Ref. 2,16], we are confident hypothesizing that the cytotoxic effect of both copolymers could be due to their ability to interact with cell membrane, thus causing direct destabilization and by enhancing permeabilization through the formation of pores.  Only subsequently to these events P5 and P7 could be internalized into the cytoplasm either through the pores or by endocytosis and act on mitochondria by stimulating ROS production thus triggering cell death. So, it is conceivable that the greater susceptibility of HTLA-ER cells, in respect to HTLA-230, could be due to changes in membrane constituents and charge occurred when they developed resistance. The membrane, becoming more negative and fluid, could allow an easier interaction and, in turn, an easier entry of the polymers into the cells. Although ROS are essential for provoking the cytotoxic effect both on HTLA-230 and HTLA-ER cells, more crucial are the electrostatic interaction of copolymers with cells surface and their entering in the cells. Probably these events are easier with HTLA-ER cells, due to occurred membrane modifications. Anyway, even if the most part of abovementioned explanation was already present in the original manuscript, further sentences to explain this question have been added in lines 768-780.

  • ROS in title should be given in full name.

In agreement with the Reviewer, we have replaced in the title ROS with Pro-Oxidant (line 3) and the full name of ROS has been inserted in the abstract (line 23). The title of the manuscript in SI has been modified accordingly.

  • A control on cytotoxicity for normal cell should provide to demonstrate the selectivity.

We agree with the Reviewer that experiments to control the behaviour of P5 and P7 toward normal cells is mandatory, to suggest our copolymers for biomedical application, but it was not in the aim of present work.

We think that, for the research to be done well and accurately, it must be done gradually. After the preparation of the new materials to be investigated as bioactive devices, and after their as complete as possible physicochemical characterization, biological investigations should firstly involve the verification of the existence of activity, then the investigation of the mechanisms of action, if possible at a molecular level, and if, as in our case, it is a matter of materials with antitumor or antibacterial activity, the verification of cytotoxicity on normal cells, that, ideally, should not be disturbed.

In this regard, in our present work, investigations have been carried out and the results have been reported and discussed, relating to the first stage, as it was in our aim. The resulting manuscript, in our opinion, is already full of interesting data and results. We therefore think that further investigations, concerning the mechanisms of action and cytotoxicity on normal cells need their own space which will be dedicated to them in future works. For the moment, as already reported in the main text (lines 798-802), we are hopeful that P5 and P7 will be not cytotoxic for normal cells according to what previously reported (Ref. 2).

  • The authors should cover the new progress of oxidation therapy in introduction for research background, but not limited to cationic polymer (e.g., Angewandte Chemie International Edition, 2020, 59(32): 13526-13530; Angewandte Chemie International Edition, 2017, 56(45): 14025-14030).

On request of Reviewer, the introduction has been enriched with reporting new progresses of oxidation therapy not limited to cationic polymers by citing the studies reported in Angewandte Chemie International Edition, 2020, 59(32): 13526-13530 and Angewandte Chemie International Edition, 2017, 56(45): 14025-14030. The references list was modified accordingly by adding the new Refs. 17 and 18. Please see lines 105-120.

  • Some small mistakes, for example, "For comparison also the cytotoxic effects of M5 and M7, the monomers from which 577 P5 and P7 were obtained, were tested on NB cells treated with increasing doses (5-250 578 µM) of them. As reported in Table 4, M5 and M7 reduced by 50 % (LD50) the viability of 579 both cell populations at the concentration of 250 μM and 100 µM respectively" here Table 4 should be Table 6.

We thank the Reviewer for his notifications and apologise for our distraction. Table 4 has been changed in Table 6.

  • "The treatments with P5 reduced cell viability of both HTLA-230 (Figure 5a, HTLA) 583 and HTLA-ER (Figure 5b, ER) in a concentration-dependent way starting from 1 µM." 1 µM should be 0.1 µM.

In this case, 1 µM is not a mistake. Indeed, we referred to the concentration from which P5 significantly reduced cell viability of NB cell, which before such concentration superimposed the viability of control cells. Anyway, for more clarity “significantly” has been included in line 651.

Reviewer 2 Report

The paper describes the synthesis of two polystyrene-based copolymers containing primary ammonium groups that carry potential as chemotherapeutic agents to combat neuroblastoma (NB). The reaction scheme via radical copolymerization, the spectroscopic characterization of the polymers, the determination of their average molar mass and of NH2 equivalents are discussed along with zeta-potential and light scattering measurements in water. The cytotoxic effect of the polymers on two human NB cell lines (one Etoposide-sensitive and one Eposide-resistant) is explored. An interesting correlation between cytotoxicity and ROS generation was demonstrated for those two NB cell lines.

The study falls within the scope of the journal and will be of interest for its readers and the wider community. My comments/suggestions are shown below:

  1. The authors should compare their work with close related systems.
  2. The authors should articulate more clearly the novelty of their work.
  3. What is the toxicity of the polymers to healthy cell?
  4. In principle, is the synthesis/purification of the polymers scalable?
  5. The conclusion section is missing.

Author Response

The paper describes the synthesis of two polystyrene-based copolymers containing primary ammonium groups that carry potential as chemotherapeutic agents to combat neuroblastoma (NB). The reaction scheme via radical copolymerization, the spectroscopic characterization of the polymers, the determination of their average molar mass and of NH2 equivalents are discussed along with zeta-potential and light scattering measurements in water. The cytotoxic effect of the polymers on two human NB cell lines (one Etoposide-sensitive and one Eposide-resistant) is explored. An interesting correlation between cytotoxicity and ROS generation was demonstrated for those two NB cell lines.

The study falls within the scope of the journal and will be of interest for its readers and the wider community. My comments/suggestions are shown below:

  1. The authors should compare their work with close related systems.

In this regard, we kindly note to the Reviewer that a comparison of results reported in our work with those of close related systems, such as the cationic peptides Lactoferrin B reported by Eliassen [2], has been already present in the non-revised form of our manuscript (713-718). We specify, that the study by Eliassen was reported because it deals with NB and therefore was considered the most suitable to make the comparisons suggested by the Reviewer.

  1. The authors should articulate more clearly the novelty of their work.

As suggested by the Reviewer, the novelty and relevance of our work has been more clearly highlighted in the new Conclusion section which was inserted as suggested by the Reviewer in the following point 5. Please, see lines 781-802.

  1. What is the toxicity of the polymers to healthy cell?

Concerning toxicity of our copolymers to healthy cells, experiments are currently underway to have such data, which we believe do not fall within the scope of this work, but require adequate space in a future work, for example dedicated to in-depth investigations to unveil the mechanisms of action of P5 and P7 at the molecular level and to evaluate their cytotoxicity towards normal cells. For the moment, as reported in the main text (lines 798-802), we are hopeful that P5 and P7 will be not cytotoxic for normal cells according to what previously reported (Ref. 2).

  1. In principle, is the synthesis/purification of the polymers scalable?

Both the synthesis and purification work-up to prepare both copolymers are simple, low-cost and easily affordable, providing polymers with percentage of conversion from good to very good and of high quality. We consider the procedures described by us scalable.

  1. The conclusion section is missing.

We kindly note the conclusion section had not been included because according to Nanomaterials instruction it is not mandatory. Anyway, since we agree with the request of the Reviewer, the conclusion section has been included as suggested. Please see lines 781-802.

Round 2

Reviewer 2 Report

The revised manuscript has been improved and is now suitable for publication.